# Multifunctional Properties of BMAP-18 and Its Aliphatic Analog against Drug-Resistant Bacteria

**DOI:** 10.3390/ph16101356

**Published:** 2023-09-25

**Authors:** Ishrat Jahan, Sukumar Dinesh Kumar, Song Yub Shin, Chul Won Lee, Sung-Heui Shin, Sungtae Yang

**Affiliations:** 1Department of Biomedical Sciences, School of Medicine, Chosun University, Gwangju 61452, Republic of Korea; upomaishrat@gmail.com; 2Department of Cellular and Molecular Medicine, School of Medicine, Chosun University, Gwangju 61452, Republic of Korea; sdkumarphd@gmail.com (S.D.K.); syshin@chosun.ac.kr (S.Y.S.); 3Department of Chemistry, Chonnam National University, Gwangju 61186, Republic of Korea; cwlee@jnu.ac.kr; 4Department of Microbiology, School of Medicine, Chosun University, Gwangju 61452, Republic of Korea

**Keywords:** antimicrobial peptide, drug-resistant bacteria, cathelicidin, multifunction, aromatic and aliphatic residue

## Abstract

BMAP-18, derived from the N-terminal region of bovine myeloid antimicrobial peptide BMAP-27, demonstrates potent antimicrobial activity without cytotoxicity. This study aimed to compare the antibacterial, antibiofilm, and anti-inflammatory properties of BMAP-18, rich in aromatic phenylalanine residues, with its aliphatic analog, BMAP-18-FL. Both aromatic BMAP-18 and aliphatic BMAP-18-FL exhibited equally potent antimicrobial activities against Gram-positive and Gram-negative bacteria, particularly methicillin-resistant *Staphylococcus aureus* (MRSA) and multidrug-resistant *Pseudomonas aeruginosa* (MDRPA). Mechanistic investigations employing SYTOX green uptake, DNA binding, and FACScan analysis revealed that both peptides acted by inducing membrane permeabilization and subsequent intracellular targeting. Moreover, both BMAP-18 and BMAP-18-FL effectively prevented biofilm formation and eradicated existing biofilms of MRSA and MDRPA. Notably, BMAP-18-FL displayed a superior anti-inflammatory activity compared to BMAP-18, significantly reducing the expression levels of pro-inflammatory cytokines in lipopolysaccharide-stimulated macrophages. This study emphasizes the similarities and differences in the antimicrobial, antibiofilm, and anti-inflammatory properties between aromatic BMAP-18 and aliphatic BMAP-18-FL, providing valuable insights for the development of multifunctional antimicrobial peptides against drug-resistant bacteria.

## 1. Introduction

Antibiotic-resistant infections have become a major global health concern, with a staggering prevalence of at least 2.8 million cases and over 35,000 annual deaths reported in the United States alone by the Centers for Disease Control and Prevention (CDC) [1,2]. The development of antibiotic resistance is attributed to several factors, including bacterial biofilm formation, enzymatic inactivation of antimicrobials, and alterations in target proteins [3,4,5]. As traditional antibiotics face mounting challenges in combating drug-resistant pathogens, there is a growing interest in exploring alternatives to antibiotics in non-compound approaches such as antimicrobial or host defense peptides, antibiofilm peptides, antibodies, vaccines, probiotics, lysins, and bacteriophages [6,7,8,9,10,11,12,13].

Amid these challenges, antimicrobial peptides (AMPs) have garnered significant interest as promising alternatives to conventional antibiotics for their multifunctional properties [14]. AMPs, also known as host defense peptides, are naturally occurring polypeptide sequences consisting of cationic and hydrophobic amino acids with direct antimicrobial activity [7,15]. AMPs can be chemically modified from their parental natural peptides based on structure activity relationships to enhance their therapeutic activity and to overcome their limitations [15,16,17]. Their rapid killing mechanism and reduced likelihood of resistance development make them particularly attractive candidates. AMPs possess key features, such as amphipathicity and net cationicity [12,18,19,20], which enable them to target anionic cell membranes and invade bacterial cytoplasm, leading to membrane permeabilization or intracellular attack, ultimately culminating in cell death [21,22,23,24,25,26,27]. Beyond their antimicrobial properties, AMPs also exhibit the ability to inhibit biofilm formation and/or eradicate existing biofilms, further adding to their potential efficacy. Moreover, AMPs demonstrate immunomodulatory effects, highlighting their multifunctional nature in the fight against drug-resistant bacteria [15]. These diverse properties render AMPs highly attractive for the development of effective antimicrobial therapeutics capable of circumventing antibiotic resistance.

One major class of AMPs, known as cathelicidins, has been identified in numerous species, including humans, sheep, cows, pigs, rabbits, and other vertebrates [28,29]. Typically, cathelicidins comprise a highly conserved N-terminal cathelin portion and an extremely variable cationic domain at the C-terminus, which plays a pivotal role in their antimicrobial function [22,30,31,32,33]. Bovine myeloid antimicrobial peptide-27 (BMAP-27), a cathelicidin peptide, adopts an α-helical structure consisting of 27 residues with an amidated C-terminus. Despite its broad-spectrum antimicrobial activity against various pathogens, the application of BMAP-27 has been hampered by its cytotoxic effects on host cells [22,28]. It is also established that the lytic effects are significantly influenced by sequence changes in neutrophil permeabilization [34]. From the previous studies, it has already been found that the removal of hydrophobic C-terminal residue from BMAP-27 or BMAP-28 lowered the cytotoxicity on mammalian cells but improved target specificity towards microbial cells [34,35]. Recently, we demonstrated that BMAP-18, a truncated version consisting of the N-terminal 18 residues of BMAP-27, possesses an amphipathic α-helical structure with clustered aromatic residues on the hydrophobic face, resulting in reduced hemolysis and improved cell selectivity compared to the full-length BMAP-27 [22]. Another study reported that shortened BMAP analogs showed notable antimicrobial activities, i.e., *P. aeruginosa*, yet it varies in different strains such as *S. aureus* [35].

In this study, we synthesized BMAP-18-FL by replacing the aromatic phenylalanine residues at four positions (F3, F6, F10, F14) of BMAP-18 with aliphatic leucine residues and compared the antibacterial, antibiofilm, and anti-inflammatory properties of aromatic BMAP-18 with its aliphatic analog, BMAP-18-FL. By highlighting the multiple functionalities of both aromatic BMAP-18 and aliphatic BMAP-18-FL, this study aims to provide significant insights into the development of promising peptides as potential alternatives to conventional antibiotics. Through a comprehensive understanding of the multifunctional properties of AMPs, we can pave the way for the development of novel, effective therapies to combat the growing threat of antimicrobial resistance.

## 2. Results

### 2.1. Peptide Design and Characterization

Table 1 presents the amino acid sequences, retention time, molecular mass, ion charge ratio, net charge, and hydrophobic moment of BMAP-18 and its aliphatic analog, BMAP-18-FL. In BMAP-18-FL, the aromatic amino acid phenylalanine (F) residues at positions F3, F6, F10, and F14 are replaced by the aliphatic amino acid leucine (L) residues. The value found from mass spectroscopy was almost identical to the theoretical value, demonstrating the accuracy of the synthetic peptides (Appendix A). The analytical RP-HPLC retention time of BMAP-18 and BMAP-18-FL was found to be 18.1 min and 20.2 min, respectively. The α-helical wheel diagram of BMAP-18 and BMAP-18-FL were predicted using the online analysis tool, HeliQuest server (https://heliquest.ipmc.cnrs.fr/cgi-bin/ComputParams.py) (accessed on 17 July 2023) (Figure 1a). The results showed that both AMPs carried a net charge of +10 with a hydrophobic moment of 0.710 and 0.693, respectively (Table 1).

The secondary structure of the peptides was assessed using CD spectroscopy in various environments: 10 mM sodium phosphate buffer (mimicking an aqueous condition), 30 mM SDS (mimicking a membrane), and 50% TFE (mimicking a hydrophobic environment) (Figure 1b). In the aqueous buffer solution, both BMAP-18 and BMAP-18-FL displayed a spectrum compatible with an unstructured peptide. However, upon exposure to SDS or TFE solutions, both peptides exhibited characteristic double-negative maxima at 208 nm and 222 nm, implying that they predominantly adopted an α-helical conformation. These findings suggest significant structural alterations during the transition from the aqueous environment to SDS or TFE conditions. Notably, BMAP-18-FL exhibited a higher degree of helicity compared to its aromatic counterpart, BMAP-18, in both the SDS and TFE environments.

### 2.2. Antimicrobial Activity

For the antimicrobial activity assay, we used two antibiotic-resistant strains: Gram-positive methicillin-resistant *Staphylococcus aureus* (CCARM 3090) and Gram-negative multidrug-resistant *Pseudomonas aeruginosa* (CCARM 2095). Using the microdilution method, we assessed the minimum inhibitory concentrations (MICs) of BMAP-18, BMAP-18-FL, and some conventional antibiotics (ciprofloxacin, oxacillin, and tetracycline). As a positive control, we included bee venom melittin for comparison. The MIC values of all tested compounds are listed in Table 2. Both BMAP-18 and BMAP-18-FL exhibited a similar antibacterial activity, with MIC values ranging from 16 to 32 µM, while the conventional antibiotics did not show any activity, even at higher concentrations (up to 1024 µM). These findings highlight the potential of the peptides as promising antimicrobial agents, particularly against the resistant strains tested.

### 2.3. Mechanism of Antibacterial Action

Mechanistic investigations were carried out using the PI uptake assay to assess the membrane disruption caused by the peptides (Figure 2a). Both resistant bacteria (MRSA and MDRPA) were stained with PI, a fluorescent dye that binds to nucleic acids in cells. For comparison, we included melittin and buforin-2, known as a membrane-lytic peptide and an intracellular non-lytic peptide, respectively. The FACScan analysis data indicated that the bacterial cells remained intact in the absence of any peptide, so no shifting has occurred (count 0%), whereas the membranes were damaged in the presence of peptides with varying percentages: BMAP-18 exhibited a greater shifting effect against the Gram-negative MDRPA (85%) than the Gram-positive MRSA (51%), indicating its strong membrane-targeting capability against the Gram-negative bacteria. On the other hand, BMAP-18-FL demonstrated a more pronounced membrane-destructive effect against the Gram-positive MRSA (86%) than the Gram-negative MDRPA (75%). These results provide compelling evidence of the membrane-targeting properties of the peptides, enabling them to disrupt the integrity of bacterial cell membranes and potentially contribute to their antimicrobial activity against the resistant strains. The differential effects on Gram-positive and Gram-negative bacteria further highlight the potential of these peptides as promising candidates for combating antibiotic-resistant pathogens.

To further elucidate the mechanism of action of the peptides, we performed the SYTOX green uptake assay of *S. aureus* (KCTC 1621). SYTOX green is known for its high affinity to DNA binding, allowing us to assess the membrane permeability of *S. aureus* in the presence of peptides (Appendix A). In comparison to melittin, both BMAP-18 and BMAP-18-FL showed no significant SYTOX green influx, similar to buforin-2. In addition, we performed a DNA binding assay of the peptides to *E. coli* bacterial plasmid pBR322 by 1% agarose gel electrophoresis (Figure 2b). Various concentrations of the peptides (ranging from 0 to 8 µM) were employed, and the electrophoretic mobility was detected using a UV illuminator. BMAP-18 exhibited binding up to 2 µM, while BMAP-18-FL showed binding up to 8 µM. This result suggests that the aromatic BMAP-18 had a higher affinity for DNA binding than the aliphatic BMAP-18-FL.

### 2.4. Antibiofilm Activity

As biofilm formation often leads to antibiotic resistance, we evaluated the antibiofilm activity of BMAP-18 and BMAP-18-FL, comparing LL-37 as a positive control (Figure 3). We measured the in vitro anti-biofilm action through the minimum biofilm inhibition concentration (Figure 3a) and minimum biofilm eradication concentration (Figure 3b). Both BMAP-18 and BMAP-18-FL demonstrated a strong inhibition of biofilm formation and the eradication of mature biofilms against MRSA and MDRPA strains. Moreover, our peptide analogs effectively inhibited up to 90% of biofilm formation starting at 16 µM (equivalent to 1 MIC). Remarkably, they showed a 90% MBEC at a low concentration of 8 µM against both strains. Overall, both BMAP-18 and BMAP-18-FL exhibited superior biofilm eradication compared to the control peptide LL-37. Due to the multifunctional properties of the cathelicidin peptide LL-37, especially for its notable antibiofilm and anti-inflammatory properties against various strains including resistant ones, we used it for comparison [36].

Furthermore, we visually confirmed the biofilm eradication by staining with crystal violet (Figure 3c,d), as well as confocal laser-scanning microscopy (CLSM) with a viability staining kit (Figure 3e,f). Biofilm-forming bacterial cells treated with peptides, dried and stained with 0.1% crystal violet. Imaging showed the concentration-dependent peptide effect visually on the mature biofilms of resistant strains. In the case of CLSM, mature biofilms were incubated in the presence or absence of peptides and then stained with a live/dead bacterial staining kit (SYTO-9/PI). Live cells were labeled with a green fluorescence dye (SYTO-9), while dead cells were identified using a red dye (PI). The higher number of dead cells confirmed the eradication activity of our synthesized peptides against the resistant bacteria (MRSA, MDRPA).

### 2.5. Cytotoxic Activity of Peptides

To evaluate the toxicity of BMAP-18 and BMAP-18-FL, we conducted hemolysis assays using sheep red blood cells (Figure 4a) and examined their cytotoxic effects on mouse macrophage RAW 264.7 cells using the MTT dye reduction assay (Figure 4b). Both peptides did not induce hemolytic activity below 16 µM. At a concentration of 64 µM, BMAP-18-FL caused less than 10% hemolysis, while BMAP-18 induced about 20% hemolysis (Figure 4a). In RAW 264.7 cells, both peptides showed a similar pattern, with over 70% cell survival (Figure 4b). Compared to melittin, both peptides demonstrated a significantly lower hemolysis of sheep red blood cells and a lower cytotoxicity against RAW 264.7 cells, suggesting their potential as safer candidates for further exploration as antimicrobial agents.

### 2.6. Anti-Inflammatory Activity

To explore the anti-inflammatory activity of BMAP-18 and BMAP-18-FL, we conducted the LPS binding assay using the BODIPY-TR cadaverine (BC) displacement assay to investigate whether their inhibitory mechanism against cytokines in LPS-stimulated RAW 264.7 cells is due to the direct neutralization of LPS. In this assay, BC fluorescence is quenched upon binding to free LPS. However, the introduction of AMPs displaces BC, resulting in the alleviation of quenching, and thus an increase in fluorescence, indicating successful binding of the compounds with LPS. As illustrated in Figure 5a, all peptides exhibited binding in a dose-dependent manner, with BMAP-18-FL demonstrating a stronger LPS-binding affinity compared to BMAP-18 or the control peptide (LL-37).

To gain insights into the mechanism of action of the peptides in macrophage cells, we pursued two approaches. First, we examined the effects of peptides on free LPS by pre-incubating the FITC-LPS/peptide mixture and then adding it to RAW 264.7 cells. As depicted in Figure 5b, BMAP-18-FL significantly suppressed the binding of FITC-LPS to macrophage cells, indicating successful peptide–LPS binding compared to its parent analog. Furthermore, we evaluated the neutralizing ability of peptides on receptor-bound LPS in RAW 264.7 cells. As shown in Figure 5c, similar to LL-37, BMAP-18-FL effectively bound to receptor-bound LPS, suggesting antagonistic activity towards LPS-binding receptors (CD14 or TLR4) by neutralizing LPS. These findings were validated by polymyxin B (PMB) as it cannot neutralize receptor-bound LPS, while LL-37 showed the opposite pattern.

We also assessed the effect of BMAP-18 peptides on the expression of proinflammatory cytokines (IL-6, MCP-1, and TNF-α) in LPS-stimulated RAW 264.7 cells using ELISA. The transcriptional expression levels of these cytokines decreased upon peptide treatment in a dose-dependent manner (Figure 5d). However, BMAP-18 could not inhibit MCP-1 expression in LPS-stimulated RAW264.7 cells, whereas BMAP-18-FL showed excellent inhibitory activities against all of these cytokines. Even at the lowest concentration of 4 µM, BMAP-18-FL demonstrated a significantly lower expression (≤10%) of cytokines, similar to the anti-inflammatory peptide LL-37.

## 3. Discussion

Antimicrobial peptides (AMPs) are an important part of the immune system and have shown promise as a first line of defense against invading pathogens [37,38]. With the increasing resistance of antibiotics, AMPs have emerged as promising candidates due to their ability to target a wide range of microorganisms, while also having a lower susceptibility to developing resistance [21,37,39]. To enhance their antimicrobial activity, AMPs require an adequate positive charge for electrostatic interactions with the bacterial membrane [37]. Additionally, a hydrophobic domain is crucial for membrane permeabilization and cytotoxicity, allowing the peptides to disrupt and penetrate the bacterial membrane effectively [16,17,40,41,42,43]. In this study, we synthesized the aromatic BMAP-18 and its aliphatic analog BMAP-18-FL to compare their antimicrobial, antibiofilm, and anti-inflammatory activities.

Previous studies have demonstrated the significance of aromatic residues, such as phenylalanine, in AMPs, as they play a crucial role in membrane disruption and antimicrobial activity owing to their hydrophobic nature and their ability to interact with bacterial membranes [44]. Conversely, aliphatic residues such as leucine may exhibit different interactions with the membrane. Considering the escalating threat of antibiotic-resistant infections caused by MRSA and MDRPA, we assessed the MIC values of our synthesized BMAP-18 and BMAP-18-FL against these two resistant pathogens. Despite the variation in amino acid composition, our findings reveal that both BMAP-18 and BMAP-18-FL exhibited comparable MIC results, ranging from 16 to 32 µM (Table 2). The outcomes from flow cytometry analysis (Figure 2a), the SYTOX green uptake assay (Appendix A), and the DNA binding assay (Figure 2b) suggest that both peptides disrupt the bacterial membrane, as well as binding to DNA. However, BMAP-18 displayed a stronger membrane-targeting capability against the Gram-negative MDRPA (85%) compared to the Gram-positive MRSA (51%), indicating its preference for Gram-negative bacteria. Conversely, BMAP-18-FL showed a more pronounced membrane-destructive effect against the Gram-positive MRSA (86%) than the Gram-negative MDRPA (75%), suggesting its preference for Gram-positive bacteria. Although there were some distinctions between BMAP-18 and BMAP-18-FL in terms of membrane permeabilization and DNA binding affinity, both peptides displayed a similar antimicrobial activity. It is also noticeable that our synthesized L-isomers of BMAP-18 analogs showing less cytotoxicity which is in agreement with previous studies: the D form of BMAP-18 was found to be more cytotoxic than its L-isomers [45,46].

Interestingly, both peptides also exhibited a strong antibiofilm activity against pre-formed biofilms of MRSA and MDRPA, further emphasizing their potential as alternative therapeutic agents against antibiotic-resistant infections. The ability to disrupt and prevent biofilm formation is crucial in combating bacterial resistance, as biofilms provide a protective environment that enables bacteria to evade conventional antibiotics. While the antimicrobial and antibiofilm activities of aromatic BMAP-18 and aliphatic BMAP-18-FL were found to be comparable, a notable difference emerged in their anti-inflammatory properties. BMAP-18-FL exhibited a superior anti-inflammatory activity compared to BMAP-18, as demonstrated by its significant reduction in the expression levels of pro-inflammatory cytokines.

Several studies have demonstrated that lipopolysaccharide (LPS), a crucial component of Gram-negative bacteria, actively controls the membrane insertion and antibacterial properties of numerous AMPs [47,48,49]. As the binding capacity of AMPs to LPS is considered a potential therapeutic target, we looked at different anti-inflammatory effects associated with LPS against our target peptides to assess its potential as a treatment for septic shock. The aliphatic BMAP-18-FL exhibited direct binding to LPS and demonstrated a strong neutralizing effect on LPS-conjugated macrophages. Initially, we evaluated the direct binding to LPS with the BODIPY-TR cadaverine (BC) displacement assay and found the dislocation of dye commenced at 1 µM (Figure 5a). Notably, BMAP-18-FL showed over two-fold (>80%) stronger binding to LPS than BMAP-18 at higher concentrations. To investigate the mechanism of LPS binding, we studied the effects of our peptides with FITC-labeled LPS on macrophage cells through flow cytometry analysis. As depicted in Figure 5b,c, the aliphatic BMAP-18-FL significantly bound to both free LPS and receptor (CD-14)-bound LPS. Our data fully complied with former studies on LL-37 that binds only with receptor-bound FITC-LPS [47,50,51,52,53].

Previous reports indicated that Toll-like receptor-4 (TLR-4), a transmembrane protein, interacts with the LPS-CD14 complex to initiate intracellular signaling, which subsequently leads to the production and release of pro-inflammatory cytokines, ultimately contributing to septic shock [47,48]. Based on the receptor binding evidence obtained from flow cytometry, we sought to evaluate the inhibitory effect of our peptides on pro-inflammatory cytokines: interleukin-6 (IL-6), monocyte chemoattractant protein-1 (MCP-1), and tumor necrosis factor-alpha (TNF-α). Our findings revealed a remarkable reduction in the levels of these cytokines with BMAP-18-FL treatment, even at lower concentrations such as 4 µM (Figure 5d). Collectively, our results suggest that BMAP-18-FL works on LPS by competitively binding to the CD-14 receptor initially and successively binds to free LPS, resulting in a significant inhibition of pro-inflammatory cytokines. These findings hold promise for the potential use of BMAP-18-FL as an effective anti-inflammatory agent in addition to its antimicrobial properties.

In conclusion, our study investigated the antimicrobial, antibiofilm, and anti-inflammatory properties of aromatic BMAP-18 and aliphatic BMAP-18-FL. Both peptides demonstrated a potent antimicrobial activity against Gram-positive MRSA and Gram-negative MDRPA. Importantly, these peptides exhibited a relatively low cytotoxicity, making them promising candidates for combating antibiotic-resistant infections. Mechanistic analysis provided valuable insights into the mode of action of both BMAP-18 and BMAP-18-FL, inducing membrane permeability and leading to intracellular targeting, which contributed to their potent antimicrobial activity. Additionally, both peptides effectively prevented biofilm formation and eradicated existing biofilms of MRSA and MDRPA. Notably, our study uncovered a distinct advantage of BMAP-18-FL over BMAP-18, as the aliphatic analog exhibited a superior anti-inflammatory activity. It significantly reduced the expression levels of pro-inflammatory cytokines in LPS-stimulated macrophages, suggesting its potential as a valuable anti-inflammatory agent alongside its antimicrobial properties. Overall, this study underscores the significance of understanding the similarities and differences between aromatic BMAP-18 and aliphatic BMAP-18-FL in their antimicrobial, antibiofilm, and anti-inflammatory properties. These multifunctional attributes of both peptides offer promising prospects for the development of novel antimicrobial therapies capable of effectively combating drug-resistant bacteria.

## 4. Materials and Methods

### 4.1. Materials

The solid-phase peptide synthesis (SPPS) materials, including the rink amide-methylbenzhydrylamine (MBHA) resin and the 9-fluorenyl-methoxycarbonyl (Fmoc) protected amino acids, were bought from Novabiochem (Darmstadt, Germany). The following substances were provided by Sigma Aldrich (St. Louis, MO, USA): lipopolysaccharide (LPS) purified from *Escherichia coli* O111:B4, FITC-labeled LPS (*E. coli* O111:B4), 2,2,2-triflluoroethanol (TFE), sodium dodecyl sulphate (SDS), 3-(4,5-dimethylthiazol-2-yl)-2,5-diphenyl-tetrazolium bromide (MTT), propidium iodide (PI). From R&D Systems (Minneapolis, MN, USA), ELISA kits for TNF-α, IL-6, and MCP-1 were purchased. Invitrogen live/dead bacterial viability kits, SYTOX green, and the *E. coli* bacterial plasmid pBR322 were purchased from Thermo Fisher Scientific, South Korea. Mouse macrophage RAW 264.7 cells were bought from the American Type Culture Collection (Manassas, VA, USA). Dulbecco’s modified Eagle’s medium (DMEM) and fetal bovine serum (FBS) were acquired from Lonza (Walkersville, MD, USA) and Atlas biologicals (Seoul, Republic of Korea), respectively. Two resistant bacteria of this study: methicillin-resistant *Staphylococcus aureus* strains (CCARM 3090) and multidrug-resistant *Pseudomonas aeruginosa* strains (CCARM 2095) were obtained from the Culture Collection of Antibiotic-Resistant Microbes (CCARM) of Seoul Women’s University in Korea. Other strains were obtained from the Korean Collection for Type Cultures (KCTC).

### 4.2. Peptide Synthesis

BMAP-18 (GRFKRFRKKFKKLFKKLS) and BMAP-18-FL (GRLKRLRKKLKKLLKKLS) were synthesized by solid-phase synthesis method using Fmoc (9-fluorenyl-methoxycarbonyl) chemistry (0.06 mmol scale) on rink amide 4-methylbenzyldrylamin (MBHA) resin followed by peptide cleavage. Both of the peptides were purified with reversed-phase high-performance liquid chromatography (RP-HPLC; Shimadzu, Kyoto, Japan) on an analytical Vydac C_18_ column (length: 250 mm, internal diameter: 20 mm, particle size: 15 μm, pore size: 300 Å) which was monitored by a UV detector at 224 nm. At first, the retention time of the peptides was observed using a small column, then a purified sample was collected from a large column with RP-HPLC. Finally, the purified products were determined by LC-MS analysis (API2000, AB SIEX, Alexandria, VA, USA).

### 4.3. Circular Dichroism Spectroscopy

The secondary structure of peptides was detected in different solutions in a J-715 CD spectropolarimeter (Jasco, Tokyo, Japan) equipped with a 1 mm path length cell. CD spectra were monitored in 10 mM sodium phosphate buffer (pH 7.4), 30 mM SDS micelles, and 50% TFE in sodium phosphate buffer between 190 nm to 250 nm in wavelength at 25 °C. The results from three scans were collected and calculated for each peptide. The CD data were expressed as the mean residue ellipticity [θ] in degree square centimeter per decimole (deg·cm^2^/dmol).

### 4.4. Antibacterial Activity

According to the guidelines of the clinical and laboratory standard institute (CLSI), the antimicrobial activity of the peptides against resistant bacteria was determined [54]. Minimum inhibitory concentrations (MICs) were measured by using the microtiter broth dilution method. All strains were cultured overnight to the stationary phase at 37 °C. The overnight cultures were 10-fold diluted in Muller–Hinton (MH) broth (Difco, Detroit, MI, USA) and incubated for a few hours to attain mid-log phase growth. The mid-log phase cultures were diluted with MHB media and added to sterile 96-well plates having two-fold serially diluted peptides and antibiotics. The plate was incubated for 18–24 h at 37 °C and the MIC was detected as the lowest concentration of peptide that prevented visible turbidity. The MIC was determined by optical density (OD) at 600 nm using a microplate ELISA (Bio-Tek Instruments EL800, San Diego, CA, USA) reader.

### 4.5. Antibiofilm Activity

Antibiofilm properties of the peptides were evaluated by measuring the minimum biofilm inhibition concentration (MBIC) and minimum biofilm eradication concentration (MBEC) against MRSA (CCARM 3090) and MDRPA (CCARM 2095). Both experiments followed previously described methods [55,56]. In the case of biofilm inhibition, a sub-culture of bacteria (~10^6^ CFU/mL) was incubated overnight at 37 °C in a 96-well plate with or without peptides. The highest concentration for each peptide was 128 μM, which was serially diluted two-fold. LL-37 was used as the control peptide and the untreated culture was considered as a negative control. The final calculation was performed after taking the absorbance reading at 600 nm and graphs were prepared by sigma plot software.

For biofilm eradication, these bacteria (~10^6^ CFU/mL) were incubated at 37 °C for 24 h, in 96-well Calgary microtiter plates containing PEG, in the absence or presence of serially diluted peptides. After the incubation, the lids were washed with PBS to remove unattached bacteria and then air-dried. Later, fixation was performed by methanol and dried wells were stained with 100 μL of 0.1% crystal violet for 5 min. After discarding the stain, 95% ethanol was added into the wells and kept in a shaker for 30 min. Finally, the absorbance was measured at 600 nm and graphs were produced by sigma plot software after calculation.

### 4.6. Confocal Laser-Scanning Microscopy

MRSA (CCARM 3090) and MDRPA (CCARM 2095) were cultured (~ 10^6^ cells/well) in 6-well plates with coverslips in MHB–glucose media to form biofilms for 48 h followed by peptide treatment for another 24 h. After that, cells were washed with PBS and a staining probe mixture (LIVE/DEAD™ BacLight™ Bacterial Viability kit; Invitrogen, Life Technologies Corporation, Eugene, OR, USA) was used according to the manufacturer’s instructions. The two nucleic acid stains of the kit, SYTO-9 and PI, were added to the cells and incubated in the dark for 30 min. The cells were then washed to remove the stain and fixation was conducted by methanol. Later, the coverslips were dried and prepared on glass slides. Biofilm without any peptide treatment was referred to as the control. Biofilm mass in coverslips were visualized using confocal laser-scanning microscopy (Zeiss Microscopy, Jena, Germany) and analyzed by ZEN 2009 Light Edition software (Zeiss Microscopy, Jena, Germany).

### 4.7. FACScan Analysis

The integrity of the bacterial cell membrane was evaluated by measuring the propidium iodide (PI) uptake by flow cytometry. In short, mid-log phase MRSA and MDPRA bacterial cells were diluted until the OD_600_ absorbance was approximately 0.5. The mixture was prepared by adding the same volume of cell suspension and PBS (1×) and centrifuged for 5 min at 8000 rpm. After discarding the PBS, cell pellets were resuspended in PBS. Then, 10 µL PI was added and incubated for 15 min. In total, 1 MIC of peptides was added to the mixture followed by incubation for another 15 min. For the control, two different types of peptides: melittin (membrane target) and buforin-2 (intracellular target) were used and the untreated culture was considered as a negative control. Finally, PI fluorescence was quantified using a FACScan instrument (Agilent, Santa Clara, CA, USA).

### 4.8. DNA Binding Assay

An agarose gel-retardation assay was carried out to determine the binding of peptides with DNA as described previously [50]. Different concentrations of peptides were mixed with 100 ng of bacterial plasmid pBR322 in a binding buffer (10 mM Tris-HCl, 5% glucose, 50 mg/mL BSA, 1 mM EDTA, and 20 mM KCl). Then, the mixture of DNA and peptide was incubated at 37 °C for 1 h followed by 1% agarose gel electrophoresis in 0.5% TBE buffer. Bands were detected using a UV illuminator (Bio-Rad, Hercules, CA, USA).

### 4.9. Hemolytic Activity

The hemolytic activity of the peptides was determined by measuring the free hemoglobin of sheep red blood cells (sRBCs). Fresh sRBCs were washed with PBS, centrifuged at 4 °C, and resuspended in PBS to make a dilution of erythrocytes (4% *v/v*). In total, 100 µL of sRBCs was suspended in a sterilized 96-well plate with 100 μL serially diluted peptides, then incubated for 1 h at 37 °C. Centrifugation was carried out for 5 min at 1000 RCF and the supernatant was transferred to a new 96-well clean plate. Using a microplate ELISA reader (Bio-Tek Instruments EL800, San Diego, CA, USA), hemoglobin release was recorded by measuring the absorbance at 405 nm. Melittin was used as a reference peptide, whereas 0.1% Triton X-100 was used for inducing 100% hemolysis. Zero hemolysis was calculated using PBS only. The percentage of hemolytic activity was calculated by the following formula: % Hemolysis = [(Abs in peptide solution- Abs in PBS]/(Abs of 0.1% Triton X-100 − Abs of PBS)] × 100.

### 4.10. Cytotoxicity

The MTT dye reduction assay was carried out against RAW 264.7 cells to determine the cytotoxicity of the peptides. Briefly, the cells were seeded in 96-well plates and incubated for 18–24 h in the presence 5% CO_2_ at 37 °C. Peptides were added in varied concentrations to the cells and incubated for 24 h followed by the addition of 20 µL MTT. After 3 h of incubation, the formed formazan crystals were dissolved in 200 µL DMSO and the absorbance was measured at 550 nm for cell viability calculation. Untreated cells were used as a negative control, whereas cytotoxic melittin was used as a positive control. Graphs were produced by sigma plot software after statistical analysis (ANOVA).

### 4.11. LPS Binding Assay

By using a BODIPY-TR cadaverine (BC) (Sigma, Darmstadt, Germany) displacement assay, the ability of peptides to bind with LPS was analyzed as in previous studies [57]. LPS from *E. coli* 0111: B4 (25 µg/mL) was incubated with BC (2.5 µg/mL) in a quartz cuvette with 50 mM Tris buffer (pH 7.4). Peptides were added in a dose-dependent manner up to 32 µM within a 60 s time interval and the fluorescence was recorded at an excitation wavelength of 580 nm and an emission wavelength of 620 nm with a spectrofluorophotometer (Shimadzu, Japan). The values were converted to %∆F using the following equation: %∆F = [(F_obs_ − F_0_)/(F_100_ − F_0_)] × 100, where F_obs_ is the observed fluorescence at a given peptide concentration, F_0_ is the initial fluorescence of BC with LPS in absence of peptide, and F_100_ is the fluorescence of BC with LPS in presence of polymyxin B which is used as a positive control.

### 4.12. Effect of Peptides on LPS Binding

The ability of peptides to bind with FITC-LPS was evaluated as in a previously described method [47]. First, FITC-LPS (1 µg/mL) from *E. coli* 0111: B4 was incubated with peptides at 4 °C for 1 h, then RAW264.7 macrophage cells (3 × 10^5^ cells/mL) were treated with the FITC-LPS/peptide mixture and incubated for 30 min at 37 °C. After that, cells were washed with PBS to remove unbound LPS and the binding affinity was measured by fluorescence intensity from a FACScan machine (Agilent, Santa Clara, CA, USA). The effect of peptides on receptor-bound LPS was assessed using another flow cytometry analysis. In brief, RAW264.7 cells were pre-incubated with FITC-LPS for 1 h in absence of peptides. After washing with PBS, the cells were incubated with peptides for 1 h followed by further washing. Finally, the binding of FITC-LPS to RAW264.7 cells was analyzed by flow cytometry. In both experiments, the LPS binding inhibitor PMB (Polymyxin B) and LL-37 were used as controls. Background fluorescence was assessed by using RAW 264.7 cells incubated without FITC-LPS or peptides.

### 4.13. Evaluation of Pro-Inflammatory Cytokines and Chemokines

To examine whether the tested peptides were able to block the inflammatory responses induced by LPS, mouse macrophage RAW 264.7 cells were stimulated with LPS (20 ng/mL) with or without peptides. Untreated cells and LPS-only stimulated cells were considered as controls. After incubation, the inhibitory effect of pro-inflammatory cytokines (IL-6, MCP-1, and TNF-α) was measured by an ELISA kit according to manufacturer’s (R&D Systems) guidelines. All samples were quantified in triplet.

### 4.14. Statistical Analysis

All graphing and statistical analysis were carried out using a one-way analysis of variance (ANOVA) with SPSS 16.0 software and Sigma plot v12.0. The data in this study were expressed as the means ± standard deviation (SD) from three independent experiments.

## Figures and Tables

**Figure 1 pharmaceuticals-16-01356-f001:**
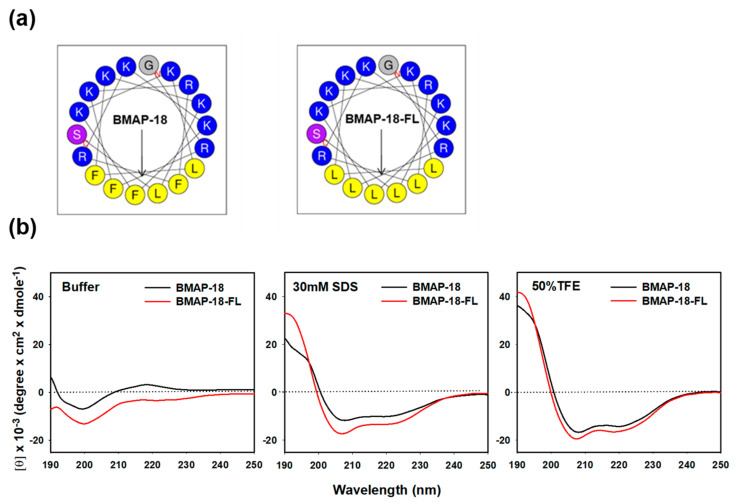
Projected structure and solution structure of the synthesized peptides: BMAP−18 and BMAP−18−FL. (**a**) Helical wheel projection diagrams of the peptides. Polar residues (positively charged) are indicated by blue circles and non-polar residues (hydrophobic region) are marked by yellow circles. The overall hydrophobic moment (µH) is represented by an arrow. (**b**) CD spectra of the peptides in aqueous and membrane-mimicking solutions: 10 mM sodium phosphate buffer, 30 mM SDS, and 50% TFE. The mean residue ellipticity was plotted against wavelength. The values from three scans were averaged per sample.

**Figure 2 pharmaceuticals-16-01356-f002:**
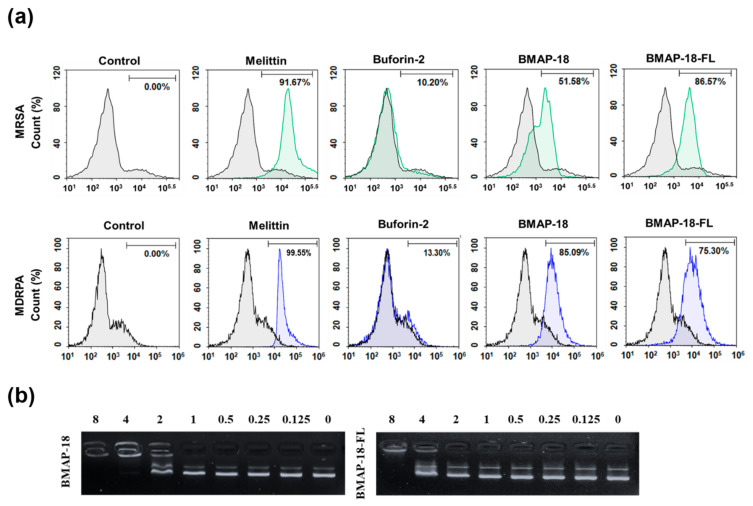
Interaction of the peptides BMAP−18 and BMAP−18−FL with membranes and intracellular DNA. (**a**) Membrane integrity disruption of MRSA (CCARM 3090) and MDRPA (CCARM 2095) in the presence of peptides (1MIC), measured by an increase in the fluorescent intensity of propidium iodide (PI). The control was processed without peptides. (**b**) Interaction of peptides with plasmid. Binding was assayed by measuring the inhibition of migration by the *E. coli* plasmid pBR322. DNA and peptides (0–8 μM) were co-incubated for 1 h at room temperature before electrophoresis on a 1% agarose gel.

**Figure 3 pharmaceuticals-16-01356-f003:**
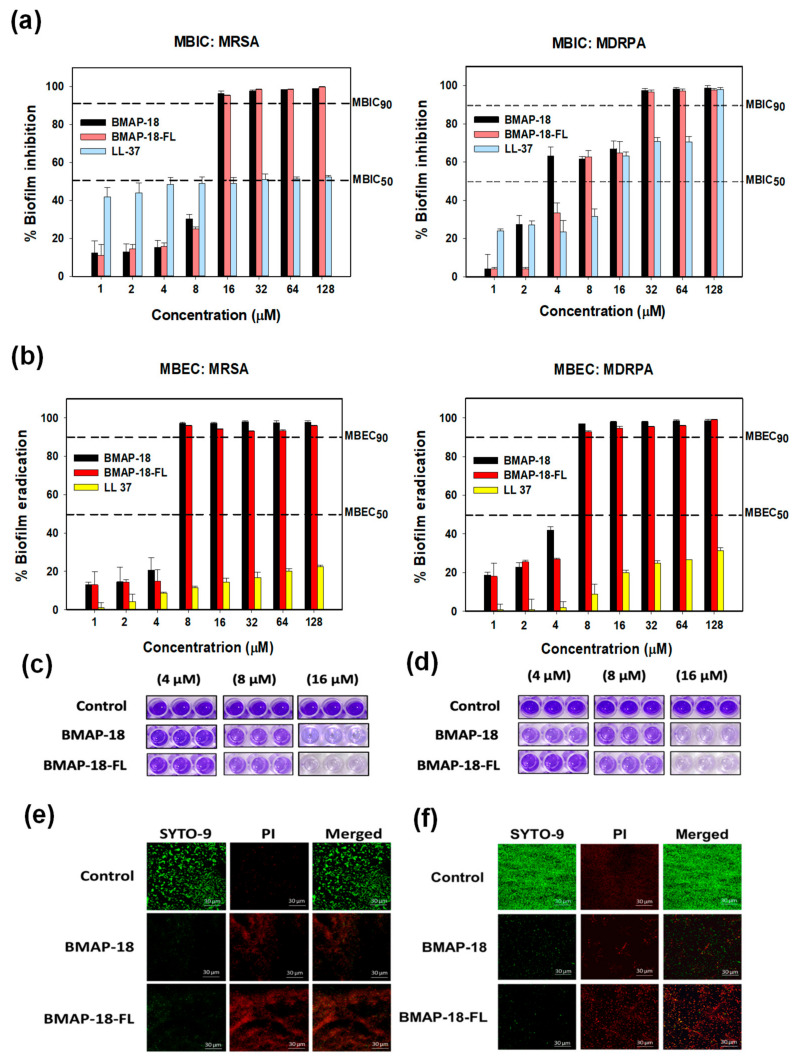
In vitro antibiofilm activities of BMAP−18 and BMAP−18−FL with reference peptide LL−37 against MRSA and MDRPA. (**a**) Biofilm inhibition (MBIC), (**b**) biofilm eradication (MBEC). Different concentrations (1–128 μM) were used to determine the MBIC and MBEC. The dashed line indicates 50% and 90% inhibition and the eradication concentration. MBEC was determined using Calgary microtiter plates containing PEG, in the absence or presence of serially diluted peptides. After washing, fixation was conducted by methanol and dried wells were stained with 100 μL of 0.1% crystal violet for 5 min. After discarding the stain, 95% ethanol was added into the wells and kept in a shaker for 30 min. Finally, the absorbance was measured at 600 nm. Graphs were produced using sigma plot software 12.0 after statistical analysis (ANOVA). MBEC images are shown from Calgary plates against MRSA (**c**) and MDRPA (**d**), respectively. Confocal laser scanning microscopy (CLSM) was used to evaluate the biofilm eradication effect of BMAP−18 and BMAP−18−FL against MRSA (**e**) and MDRPA (**f**). Biofilms were visualized with a live/dead bacterial viability staining kit (SYTO 9/PI). Live and dead cells are indicated by green fluorescence (SYTO 9) and red fluorescence (PI), respectively. The scale bar was 30 µM.

**Figure 4 pharmaceuticals-16-01356-f004:**
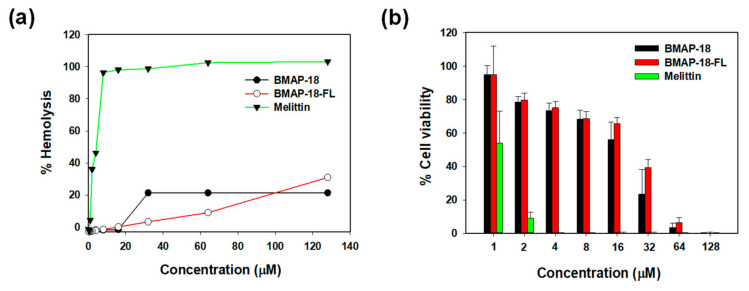
Cytotoxic activities of BMAP−18 and BMAP−18−FL with the control peptide melittin. (**a**) Hemolytic activity of the peptides, measured as the percentage hemolysis in sheep red blood cells (sRBCs). (**b**) Cytotoxicity of the peptides, measured as the percentage of cell viability in RAW264.7 mouse macrophages using an MTT dye reduction assay. Graphs were prepared using sigma plot software after statistical analysis (ANOVA).

**Figure 5 pharmaceuticals-16-01356-f005:**
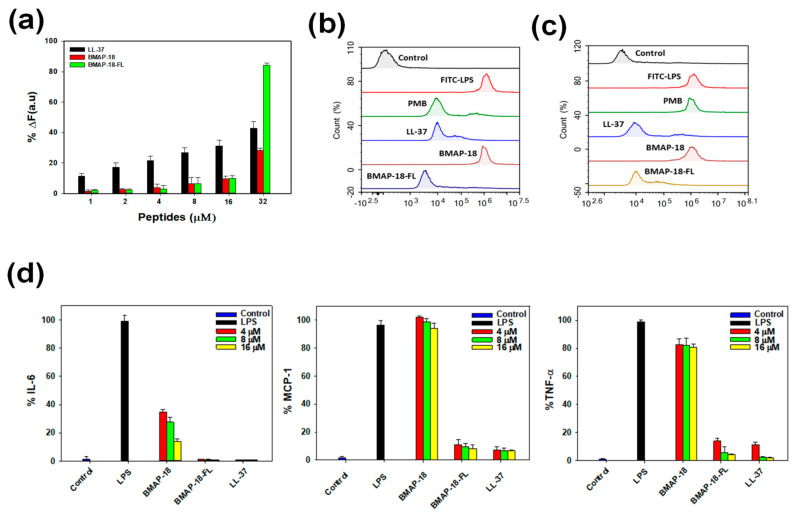
Anti-inflammatory activity of BMAP−18 and BMAP−18−FL with reference peptide LL−37. (**a**) Peptide binding affinity to LPS from *E. coli* 0111: B4 using the BODIPY−TR cadaverine (BC) displacement assay. The fluorescence was recorded at an excitation wavelength of 580 nm and an emission wavelength of 620 nm. (**b**) FACScan analysis of mouse macrophage RAW 264.7 cells treated with FITC−LPS (1 µg/mL) from *E. coli* 0111: B4. FITC−LPS was pre-incubated with individual peptides for 1 h and then RAW264.7 cells were treated with each peptide/LPS mixture. (**c**) RAW264.7 cells were pre-incubated with FITC−LPS for 1 h in absence of peptides. The cells were then treated with peptides for an additional 1 h. The control was recorded from the fluorescence value of untreated cells. Polymyxin B and LL−37 were used as reference peptides. Values given with the peaks represent the percentage of fluorescence intensity. (**d**) Effects of BMAP−18 and BMAP−18−FL with the control peptide LL−37 on IL−6, MCP−1, and TNF−α in LPS-stimulated macrophage RAW264.7 cells. The concentrations of peptides used for this experiment were 4–16 μM. All data represent at least three independent experiments and are expressed as the mean  ±  standard error of the mean (SEM). Graphs were produced using sigma plot software after statistical analysis (ANOVA).

**Table 1 pharmaceuticals-16-01356-t001:** Amino acid sequences and physicochemical properties of BMAP-18 and BMAP-18-FL.

Peptides	Amino Acid Sequence	t_R_ (min) ^a^	Molecular Mass (g/mol)	MS Analysis ^b^			Net Charge	Hydrophobic Moment (µH) ^c^
				Z	*m*/*z* Calculated	*m*/*z* Found		
BMAP-18	GRFKRFRKKFKKLFKKLS	18.1	2342.92	[M + 4H]^4+^	585.73	586.45	+10	0.710
BMAP-18-FL	GRLKRLRKKLKKLLKKLS	20.2	2206.85	[M + 4H]^4+^	551.71	552.40	+10	0.693

^a^ Retention time (t_R_) was measured by analytical RP-HPLC with C_18_ column, ^b^ Molecular masses were determined by electrospray ionization mass spectrometry (ESI-MS). Z: ion charge, *m/z*: mass-to-charge ratio, ^c^ Hydrophobic moment (μH) was calculated online at: http://heliquest.ipmc.cnrs.fr/cgi-bin/ComputParams.py (accessed on 17 July 2023).

**Table 2 pharmaceuticals-16-01356-t002:** Minimum inhibitory concentration (MIC: µM) * of BMAP-18 peptides and common antibiotics against resistant strains.

	Minimum Inhibitory Concentrations (MICs:µM) *					
Strains	Ciprofloxacin	Oxacillin	Tetracycline	BMAP-18	BMAP-18FL	Melittin
MRSA (CCARM 3090)	>1024	>1024	>1024	16	16	4
MDRPA (CCARM 2095)	>1024	>1024	>1024	32	32	8

* Minimum inhibitory concentrations (MICs) were determined as the lowest concentration of the antimicrobial agent that inhibited bacterial growth.

## Data Availability

Data is contained within the article.

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
