# Peer review of "Multifunctional Properties of BMAP-18 and Its Aliphatic Analog against Drug-Resistant Bacteria"

_pharmaceuticals, 2023, doi:10.3390/ph16101356_

Round 1
Reviewer 1 Report
The authors explored the effect of substituting F with L in the N-terminal fragment of the bovine cathelicidin BMAP27. The novelty of these kind of works is limited, as similar projects have been already performed several times with numerous other peptides. However, the authors are interested in investigating the results of a precise intervention on the sequence of the peptide and start from a reasonable rationale.
The paper is overall well written and covers several aspects of the in vivo characterization of a new compound. Moreover, authors perform all the assays using the native peptide as a control, which is mandatory for these kind of characterization.
However, some critical aspects need to be fixed before publication.
As first, the authors did not take in account several paper published along the years on the BMAP 18 compound, and as they are currently presented, some results seem findings of the current work, when in some cases represent well known phenomena. Long story short, authors presented this paper like an island on the sea, when there is a background that must be taken in account both in the introduction section and in the discussion. For example, indications of permeabilizign activity of Bmap18 [referred as BMAP27(1-18)]have been already reported by Skerlavaj et al already in 1996 on Journal of biological chemistry. These preexisting data must be considered and discussed with data produced in the current work. Similarly, the antimicrobial activity toward of Bmap18 antibiotic-resistant S. aureus were reported by Mardirossian et al in 2016 on Amino Acids. Indications of cell suffering and hemolysis induced by Bmap18 were also already reported in Skerlavaj 1996 JBC, as well as MTT viability assays, showed by Mardirossian et al 2017 on Frontiers in Chemistry. Moreover, Degasperi et al in 2020 used D-BMAP18 to evaluate its anti-inflammatory properties. Since the authors used the L-BMAP18, a comparison and discussion of these data would be desirable.
Secondarily, in Paragraph 2.2. Authors draw conclusion that are not supported by the presented data. This must be modified before publication.
Authors can not “confirm intracellular target mechanism” with a similar experiment. Co-incubation of cationic AMPs and DNA will obviously end in an interaction mainly due to electrostatic interaction. Moreover, the conditions are not really representative of what happens in vivo in a bacterial cells. Authors must therefore tune down this statement. In fact, although the different binding capacity between BMAP18 and BMAP18FL is convincing, on the basis of the current data authors can only suggest a possible interaction of these peptides with DNA (and RNA too?) in vivo and hypothesise a role in the mode of action of these compound, maybe as “sand in a gearbox” model, therefore with non-specific but effective disturb by the peptide of essential intracellular mechanisms.
This is also true for the discussion (pag 9 line 284-5). Do not state as “demonstrated” the engagement with intracellular targets.
MINOR POINTS:
1. Pag 3 line 97: rephrase as “displayed a spectrum compatible with an unstructured peptide”
2. Pag 4 line 113: add “…antibiotic-resistant strains…”
3. Pag 4 line 136: it is improper to talk about a generic “shifting” without specifing the values that are shifted.. Similarly, authors show percentages but do not refer them to the bacterial population. More detailed and specific terms are needed to guide any reader that is not confident with flow cytometry.
4. Paragraph 2.2. Authors used propidium iodide for the 2 main strains and sytox greeen for the additional one, then placed in supplementary without any specification about methods (that looks like simple fluorescence measurement), while in the dext it may sounds described like flow-cytometry. Authors should specify this point, or erase these additional results from the paper.
5. Pag 5 line 177: why authors decided to choose LL-37 as comparison?
6. Pag 10, line 381: 30 mM SDS and 50% TFE where in water or in 10 mM sodium phosphate buffer?
7. Pag11 line 401: provide in this paragraph at least some details of the methods described in ref. 45 and 46. Otherwise the reader is forced to go throug other papers to get even basic informations.
8. Pag 7 line 237. Typo in BMAP.
The quality of english is good for publication.
Reviewer 2 Report
The manuscript entitled "Enhancing Therapeutic Efficacy of Antimicrobial Peptides through Conjugation" provides a comprehensive overview of structural modification strategies pertaining to the conjugation of AMPs and their potential modes of action. However, there are some significant areas that require attention.
Feedback:
1. In line 38, it would be advantageous to introduce antimicrobial peptides and their modifications more informatively. Relevant sources that could enhance this introduction are "The Lancet Infectious Diseases," Volume 16, Issue 2, Pages 239 – 251, DOI: https://doi.org/10.1016/S1473-3099(15)00466-1; and "Chemical Society Reviews," 2021, Volume 50, Pages 4932-4973, DOI: https://doi.org/10.1039/D0CS01026J.
2. On line 270, the hydrophobic effect's role in the antibacterial activity and cytotoxicity of antimicrobial peptides is pivotal. This aspect has been investigated by multiple research groups, including those mentioned in "Biochimica et Biophysica Acta, Biomembranes," 2015, Volume 1848, Pages 2351–2364; "Biochimica et Biophysica Acta (BBA) - Biomembranes," 2020, Volume 1862, Issue 8, Pages 183195, DOI: https://doi.org/10.1016/j.bbamem.2020.183195; and "Aggregate," 2023, Pages e329, DOI: 10.1002/agt2.329.
3. Line 374 lacks the characterization of peptides through HPLC (High-Performance Liquid Chromatography). This characterization step is currently missing from the manuscript and should be addressed.
Reviewer 3 Report
Jahan et al. investigated the antibacterial, antibiofilm, and anti-inflammatory properties of BMAP-18 and its aliphatic analogue, BMAP-18-FL. They performed a lot of assays and got some data on their molecular function. However, most of these data had been reported in other studies, and some conclusions were controversial to other’s studies even the authors’ previous studies. Most of my concerns on this manuscript are listed bellow:
1. The antibacterial activity, structural properties (CD spectrum) and LPS-neutralizing activity (anti-inflammatory properties) of BMAP-18 and BMAP-18-FL has been included in the authors’ previous studies (Peptides 2011, 32, 1123–1130; Peptides. 2019, 118:170106.).
2. The authors claimed that BMAP-18 exerted only antibacterial activity, while BMAP-27 showed potent activity against bacteria in their previous studies (Peptides. 2019, 118:170106.). This is in line with some reports from other studies that the antibacterial activity of BMAP-18 was less active, especially in vivo. However, in this study, the authors made a contrary conclusion that BMAP-18 demonstrated potent antimicrobial activity without cytotoxicity.
I'd strongly recommend that the authors focus on the new discoveries, remove the repeated data and reorganize the manuscript.
Generally, the language of this manuscript was fluent with certain readability. However, there were a few grammar problems, and some sentences are confusing in logicality.
Reviewer 4 Report
Dear authors, please attend to the following observations
1. Correctly define positive and negative controls for each test performed
2. The paragraphs are very long (274-304, 305-331,332-349), divide the paragraphs to make the ideas more understandable
3. Describe the materials as the methodology is described
4. The methodology describes a statistical analysis, which is not reflected in the tables and graphs
5. How was it determined statistically which of the evaluated treatments presented the best effect?
6. The results are not adequately discussed, the discussion should not be a repetition or ratification of the results and should be based on an exhaustive review of the existing literature on the subject.
7. The conclusion should be written based on the significance of the results obtained and the aim
Round 2
Reviewer 1 Report
The authors have modified the paper which is now more interconnected with the letterature regarding the BMAP18 and is well located in the scientific landscape of cathelicidins.
However, a single point still needs to be fixed both in the results and in the discussion sections:
Pag. 4 line 165
Pag. 9 line 293-295
The authors can not state that BMAP18 has intra-bacterial targets only because of an interaction between a cathionic peptide and the negatively charged DNA, that may be simply due to electrostatic interaction, especially if observed in vitro under very simplified conditions.
Authors can only suppose that a binding of DNA may be part of a putative secondary non lytic mechanism of action.
To demonstrate a relevant intra-cellular target as a part of a primary killing mechanism, authors should find evidences of binding of the peptide to DNA in vitro under non-lytic concentrations of peptide.
Reviewer 3 Report
I tried to pick up the new elements in this manuscript i.e. antibiofilm, DNA binding, and targeting antibiotic-resistant strains. They can be reorganized to a more interest-focused research manuscript. The authors have performed a lot of comparisons in their previous studies. They have compared the structure, cell selectivity, hemolytic activity, antimicrobial activity, membrane disruption, anti-inflammatory and so on among these peptides. A more comprehensive comparison could not be a competent presentation for their new discoveries, and if they insist this, a more defined explanation should be made that what’s the conclusions from previous comparison and what’s the harvest by new comparison. Especially, one of the most important conclusions listed in the abstract “BMAP-18-FL displayed superior anti-inflammatory activity compared to BMAP-18” was also mentioned in their previous studies where a more detailed conclusion was made that “The order of inhibitory activity to inflammation by the inhibition of NO production and TNF release follows BMAP-27 > BMAP-18-W> BMAP-18-f > BMAP-18-L, BMAP-18-I, BMAP-18”. Another comparison inevitably covered the repeated works and data. I consider it inappropriate in a non-review paper.
Reviewer 4 Report
Dear authors:
In the methodology, clarify which activity was determined, antimicrobial or antibacterial?
Round 3
Reviewer 3 Report
I basically agree to the publication of this manuscript .